# The Toll pathway inhibits tissue growth and regulates cell fitness in an infection-dependent manner

Federico Germani[1†], Daniel Hain[2†], Denise Sternlicht[1], Eduardo Moreno[2,3]*, Konrad Basler[1]*

[1]Institute of Molecular Life Sciences, University of Zurich, Zurich, Switzerland; [2]Institute of Cell Biology, University of Bern, Bern, Switzerland; [3]Champalimaud Research Center Lisbon, Lisboa, Portugal

**Abstract** The Toll pathway regulates the cellular response to infection via the transcriptional upregulation of antimicrobial peptides. In *Drosophila*, apart from its role in innate immunity, this pathway has also been reported to be important for the elimination of loser cells in a process referred to as cell competition, which can be locally triggered by secreted factors released from winner cells. In this work, we provide evidence that the inhibition of Toll signaling not only increases the fitness of loser cells, but also bestows a clonal growth advantage on wild-type cells. We further demonstrate that this growth advantage depends on basal infection levels since it is no longer present under axenic conditions but exacerbated upon intense pathogen exposure. Thus, the Toll pathway functions as a fine-tuned pro-apoptotic and anti-proliferative regulator, underlining the existence of a trade-off between innate immunity and growth during development.
DOI: https://doi.org/10.7554/eLife.39939.001

*For correspondence:
eduardo.moreno@research.
fchampalimaud.org (EM);
kb@imls.uzh.ch (KB)

†These authors contributed equally to this work

Competing interests: The authors declare that no competing interests exist.

## Introduction

The Toll pathway was first identified as a master regulator of dorso-ventral patterning in the *Drosophila* embryo (*Anderson et al., 1985*) and subsequently shown to serve as the core mechanism for the innate immune response to Gram positive bacteria, fungi, viruses and cancer cells in both *Drosophila* and mammals (*Lemaitre et al., 1996*; *Medzhitov et al., 1997*; *Krutzik et al., 2001*; *Silverman and Maniatis, 2001*; *Thoma-Uszynski et al., 2001*; *Hoffmann and Reichhart, 2002*). In *Drosophila* the Toll pathway is activated upon infection via the proteolytic cleavage of the cytokine Spätzle (*Chasan and Anderson, 1989*; *Weber et al., 2003*; *Stein et al., 1991*), which triggers a signal transduction cascade that leads to the nuclear translocation of the NFκB transcription factors Dorsal and Dif (*Drier et al., 1999*). This ancient and highly conserved cascade starts with the binding of cleaved Spätzle to one of the nine Toll receptors, which in turn associates via its cytoplasmic domain with Myd88, Tube and Pelle. The kinase Pelle then phosphorylates Cactus (IκB), targeting it for proteosomal degradation (*Horng and Medzhitov, 2001*; *Sun et al., 2002*; *Wu and Anderson, 1998*). Since Cactus normally retains Dorsal and Dif in the cytoplasm, its degradation causes their release, nuclear translocation and expression of antimicrobial peptides (AMPs), molecules that specifically fight infection (*Bulet et al., 1999*).

The Toll pathway has more recently been proposed to mediate the elimination of unfit cells (*Meyer et al., 2014*) from tissues via a process known in *Drosophila* and mammals as cell competition (*Morata and Ripoll, 1975*). Weakened, damaged cells, referred to as 'losers' in the context of cell competition, are detected and eliminated from developing tissues via delamination and apoptosis when surrounded by fitter cells, generally referred to as 'winners'. Two well studied paradigms of cell competition are: *Minute* competition, where cells lacking one allele of a ribosomal protein gene

are surrounded by wild-type cells (*Morata and Ripoll, 1975*), and super-competition, where wild-type cells are surrounded by cells expressing elevated levels of *Myc* (*Moreno and Basler, 2004*; *de la Cova et al., 2004*). Overexpression (OE) of Cactus in loser cells rescues cell competition-driven elimination of both *RpL14$^{+/-}$Minute* clones and of wild-type clones surrounded by cells with an extra copy of the *dMyc* gene (*Meyer et al., 2014*). It has recently been proposed that dMyc-expressing winner cells release Spätzle and the serine proteases required for its activation, thereby enhancing Toll-dependent apoptosis in neighboring loser cells (*Alpar et al., 2018*).

Unexpectedly, we observed that inhibition of the Toll pathway by Cactus OE not only rescued the elimination of loser cells (*Figure 1—figure supplement 1*), but also conferred a clonal growth advantage to wild-type cells (*Figure 1*). Moreover, the activation of the pathway via Toll-7 or Pelle overexpression caused a reduction of clonal growth: such clones assumed a rounded-up shape, suggesting that cells were undergoing apoptosis (*Figure 1A–1A'''*, quantified in 1B). Additional regimes of clonal induction (*Figure 1C–D*) and the use of other pathway components (*Figure 1—figure supplements 2* and *3*) corroborated these results: also overexpression of Dorsal, Tollo and Toll-2 led to a reduction of clonal growth, as well as the overexpression of Toll-10b, the constitutive active form of the Toll-1 receptor. Next, we checked whether the reduction of growth observed upon Toll pathway activation was due to increased apoptosis. Dcp-1 staining revealed that both Toll-7 and Pelle OE clones are highly apoptotic, round up and are finally pushed out of the tissue via delamination (*Figure 1—figure supplement 4*). We further overexpressed Toll-7 and Cactus in the posterior compartment of the wing disc. Toll-7 OE massively induced apoptosis, whereas Cactus OE caused a reduction of apoptosis when compared to the wild-type anterior compartment (*Figure 1—figure supplement 5*).

In order to determine whether the Toll pathway regulates growth systemically, we analyzed entire animals or whole appendages with altered pathway activity (*Figure 1—figure supplement 6*). However, no major size differences could be observed, indicating that Toll signaling influences cellular fitness only in situations where cell populations that differ in their ability to respond to infection are intermingled. A corollary of this hypothesis would be that wild-type cells must exhibit higher pathway activity compared to Cactus OE cells, in which the pathway is effectively blocked. Since Toll signaling is a key mediator for the immune response, it is possible that a subliminal pathway activity level stems from chronic exposure to pathogens, due to typically unsterile working conditions. To test this, we grew animals in a fully axenic environment – that is in the complete absence of pathogens - and in a highly infected environment – that is by adding *Aspergillus niger* to the culture medium (*Figure 2—figure supplement 1*). This fungus was chosen because of its pathogenic activity that shortens the life span of flies (*Figure 2—figure supplement 2*).

As expected, in a *Minute* cell competition scenario, the inhibition of the Toll pathway via overexpression of Cactus in loser cells strongly rescued their elimination in both normal and infected conditions (*Figure 2A'–A'', B'–B''*, quantified in *Figure 2E, F*). However, no rescue of cell competition-driven elimination of loser cells was observed when animals were grown under axenic conditions (*Figure 2A,B*, quantified in *Figure 2E, F*). Indeed, under axenic conditions Cactus OE *Minute* loser clones were eliminated even more efficiently than control *Minute* clones (*Figure 2A,B,C–D'*, quantified in *Figure 2E, F*). A similar behavior was observed in the dMyc super-competition context, where Cactus OE rescued loser cell elimination upon infection but not under axenic conditions (*Figure 2G*). Thus the involvement of the Toll pathway in cell competition depends on the presence of pathogens in the environment in which larvae develop.

The clonal growth influence by Cactus OE in wild-type cells also depends on the state of infection. Under extra-pathogen conditions Cactus-dependent overgrowth was further enhanced, but suppressed under axenic conditions (*Figure 3A–B''*, quantified in *Figure 3D*). The opposite effect could be observed when clones overexpress Toll-7: clones were rare and small in both normal and axenic conditions, but grew almost as well as control *LacZ* clones upon highly infected conditions (*Figure 3C–C''*, quantified in *Figure 3D*). Thus the clonal disadvantage caused by Toll pathway activation inversely correlates with the level of infection, substantially diminishing with increasing degrees of infection. Finally, we asked whether the partially lethal effects of ubiquitous Toll-7 or Cactus OE (*Figure 1—figure supplement 2*) depend on the pathogen load (infection or axenic conditions). We found that pathway activation-induced lethality was ameliorated under conditions of extra-infection, while pathway suppression-induced lethality was rescued by axenic conditions (*Figure 3—figure supplement 1*).

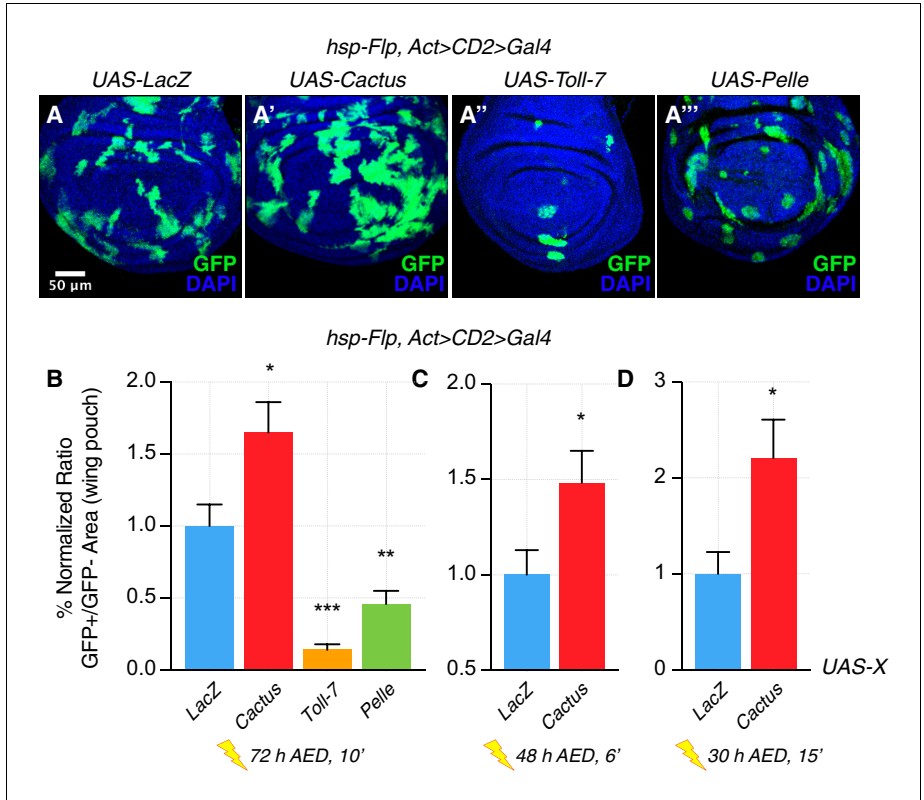

**Figure 1.** The Toll pathway negatively regulates clonal growth. When compared to LacZ overexpressing (OE) clones, here used as a control (**A**), clonal inhibition of the Toll pathway via the overexpression of the NFkB inhibitor *Cactus* (IkB) causes overgrowth (**A'**). Pathway activation via the overexpression of *Toll-7* (**A''**) or *Pelle* (**A'''**) causes growth reduction. Clones are induced 72 hr AED with a 10' heat shock. Data are quantified as the percentage of the normalized ratio between GFP$^+$ and GFP$^-$ tissue areas in the wing pouch (**B**). Similarly, Cactus OE clones grow larger when the heat shock is performed at different developmental stages and for different durations, respectively for 6' 48 hr AED (**C**) and for 15' 30 hr AED (**D**). ***p<0.001, **p<0.01, *p<0.05, t-test. Bars represent SEM.

DOI: https://doi.org/10.7554/eLife.39939.002

The following figure supplements are available for figure 1:

**Figure supplement 1.** Toll pathway inhibition rescues cell competition-driven elimination of *minute* clones.
DOI: https://doi.org/10.7554/eLife.39939.003
**Figure supplement 2.** Viability assay identifies lethal and partially lethal Toll pathway alterations.
DOI: https://doi.org/10.7554/eLife.39939.004
**Figure supplement 3.** The Toll pathway negatively regulates growth.
DOI: https://doi.org/10.7554/eLife.39939.005
**Figure supplement 4.** The Toll pathway induces delamination and apoptosis.
DOI: https://doi.org/10.7554/eLife.39939.006
**Figure supplement 5.** Compartmentally induced Toll pathway causes apoptosis.
DOI: https://doi.org/10.7554/eLife.39939.007
**Figure supplement 6.** Toll pathway modifications have no or little effect on organismal and organ growth.
DOI: https://doi.org/10.7554/eLife.39939.008

In conclusion, our findings reveal that in a non-sterile environment cells deficient for the Toll-mediated immune response grow better than immunocompetent wild-type cells. This growth difference depends on the level of infection: it is null in axenic conditions and enhanced by addition of pathogens. Most likely, therefore, it is driven by different levels of Toll pathway activity. Importantly though, these different levels must occur in cell populations that cohabitate in the same tissue, as we were not able to detect growth effects in organs entirely programmed to exhibit elevated or reduced Toll signaling. Our findings can therefore be explained by a model (*Figure 4*) in which cells

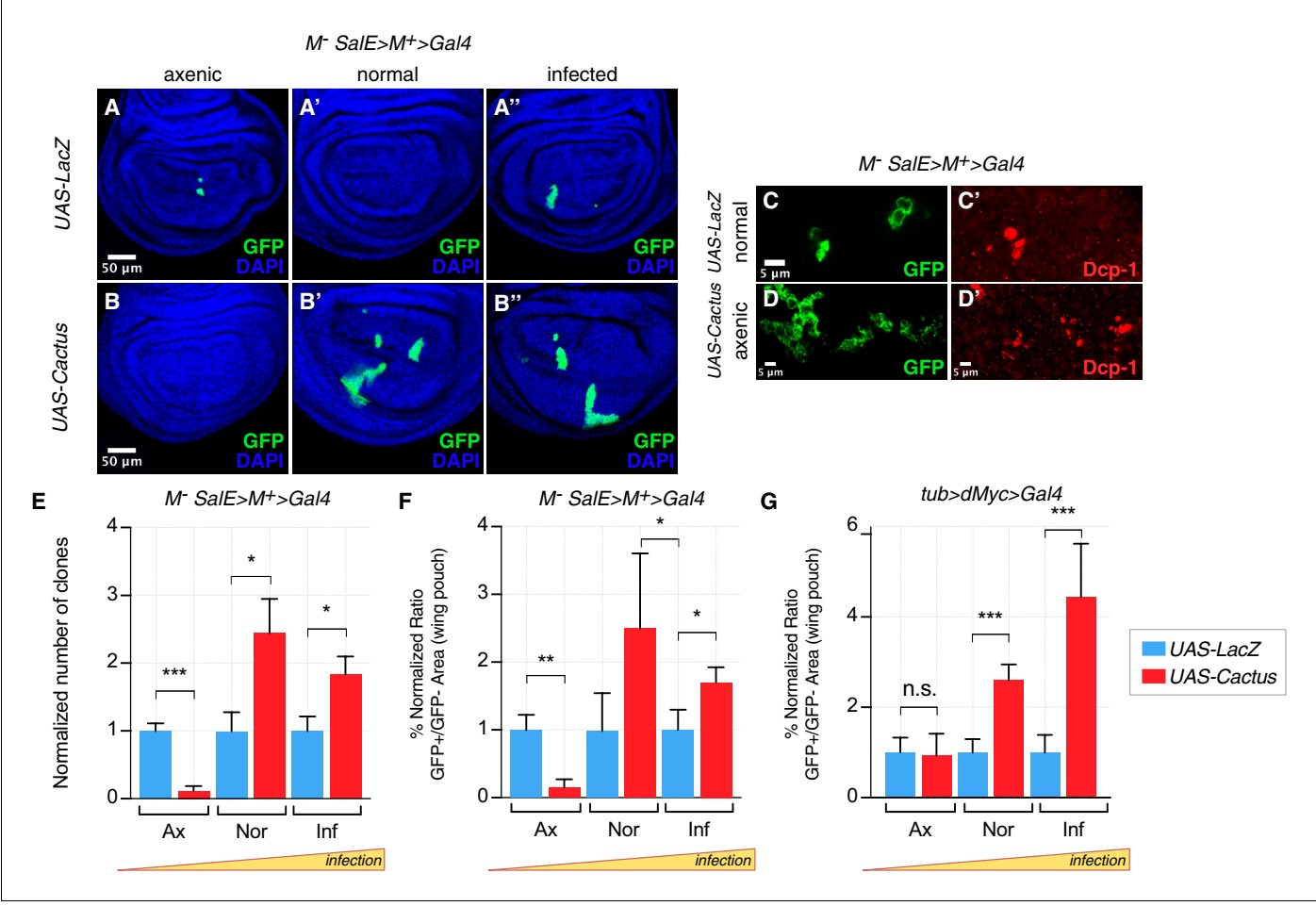

**Figure 2.** Toll pathway inhibition rescues cell competition-driven elimination of loser clones in an infection-dependent manner. *Minute* clones, which lack a functional copy of the *RpL14* gene, are eliminated via cell competition under axenic (A), normal (A'), and infected conditions (A''). LacZ overexpression in loser cells is used as a negative control. Toll pathway inhibition via Cactus OE fails to rescue cell competition-driven elimination of *minute* clones under axenic conditions (B), whereas elimination is rescued under both normal (B') and infected (B'') conditions. Loser clones are highly apoptotic, as shown with a Dcp-1 staining (C, C'). and Cactus OE escaper clones under axenic conditions are extremely fragmented and apoptotic (Dcp-1 staining, (D, D') Data are quantified by scoring the number of clones in the wing pouch (E), and by calculating the percentage of the normalized ratio between GFP$^+$ and GFP$^-$ tissue areas in the wing pouch (F). Similarly, the elimination of loser clones with a single copy of dMyc in a background with two copies of dMyc is rescued specifically under normal or, even more efficiently, under infected conditions, but not under axenic conditions, as quantified in (G). ***p<0.001, **p<0.01, *p<0.05, n.s. not significant, Mann-Whitney test. Bars represent SEM.

DOI: https://doi.org/10.7554/eLife.39939.009

The following figure supplements are available for figure 2:

**Figure supplement 1.** Methods to grow axenic or infected animals.
DOI: https://doi.org/10.7554/eLife.39939.010

**Figure supplement 2.** Flies infected with *A.niger* have a reduced life span.
DOI: https://doi.org/10.7554/eLife.39939.011

with lower Toll pathway activity profit from, and grow faster than, nearby cells with higher activity. Conversely, cell clones with higher Toll signaling levels (e.g. by overexpressing Toll receptors) are eliminated from the tissue via apoptosis and delamination when surrounded by cells with lower levels (*Figure 4—figure supplement 1*).

Our experiments indirectly suggest that the local effects on clonal growth depend on a systemic response to infection. Likely therefore, the active form of Spätzle is produced in distant organs and reaches the wing disc through the haemolymph. However, it has recently been proposed that wing discs locally produce Spätzle and the serine proteases responsible for its activation (*Alpar et al., 2018*). Via an increased secretion of these factors, Myc OE cells may be able to induce Toll-

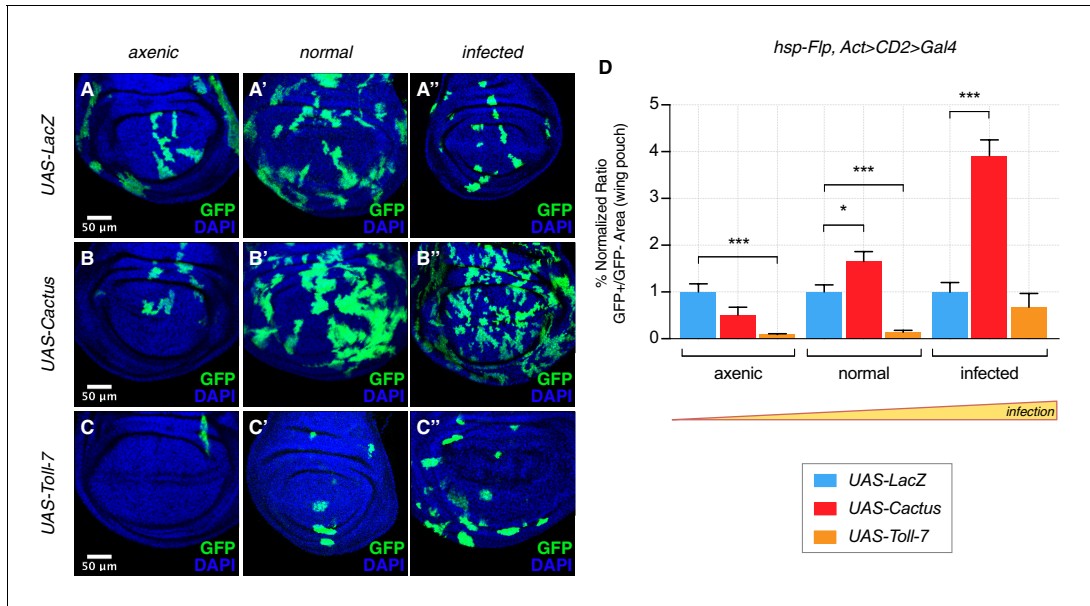

**Figure 3.** The Toll pathway negatively regulates clonal growth in an infection-dependent manner. Under normal laboratory conditions, when compared to LacZ OE clones (A'), clonal inhibition of the Toll pathway with Cactus OE causes overgrowth in a wild-type background (B'). When the pathway is activated via the overexpression of Toll-7, clones are instead reduced in size (C'). The growth advantage provided by the inhibition of the pathway is further enlarged under infected conditions (B'') compared to (A''). Inhibition of the pathway under axenic conditions does not provide a growth advantage over the surrounding cells (B) compared to (A). On the contrary, pathway activation enhances the growth-defective phenotype in axenic conditions (C) and the effect is no longer evident when larvae develop under infected conditions (C''). Data are quantified in (D) as the percentage of the normalized ratio between GFP$^+$ and GFP$^-$ tissue areas in the wing pouch. ***p<0.001, *p<0.05, Mann-Whitney test. Bars represent SEM.
DOI: https://doi.org/10.7554/eLife.39939.012

The following figure supplement is available for figure 3:

**Figure supplement 1.** Pathway activation-induced lethality is partially rescued under infected conditions and pathway suppression-induced lethality is rescued under axenic conditions.
DOI: https://doi.org/10.7554/eLife.39939.013

dependent apoptosis in neighboring loser cells (*Alpar et al., 2018*). It is therefore possible that local and systemic sources of Spätzle co-exist. Since infection is the initial trigger for the aforementioned effects in cell competition and clonal growth, it will be interesting to investigate whether the local production of Spätzle and serine proteases depends on a systemic response to infection.

Our findings also underline the important awareness that we are working in non-anthropic environments. Experimental results can dramatically differ because of complex and often unexpected consequences of immune responses. This aspect has been emphasized with experiments conducted in mice in recent years (*Beura et al., 2016*; *Abolins et al., 2017*; *Willyard, 2018*). Here we show that analogous issues can affect *Drosophila* research.

Finally, our results also reveal a phenotypic connection between the two fundamental processes of innate immunity and cell growth at the cellular level. The previous findings that Dorsal induces the transcription of the pro-apoptotic gene *rpr* (*Meyer et al., 2014)* and that the *Drosophila* Toll pathway cross-talks with the growth controlling Hippo signaling pathway (*Liu et al., 2016*) also suggest a potential mechanistic connection for our observations. The evolutionary implications can be viewed in the light of the life history theory, which seeks to explain natural selection on the basic assumption that environmental resources are limited and organisms establish trade-offs between processes such as reproduction, growth and immunity (*Roff, 1993*; *Stearns et al., 2000*). Allocating resources into a costly trait like immunity (*van der Most et al., 2011*; *Rauw, 2012*) occurs at the expense of other important processes, such as organismal growth. In agreement with this theory, we experimentally show that a cellular trade-off exists between innate immunity and clonal growth during development.

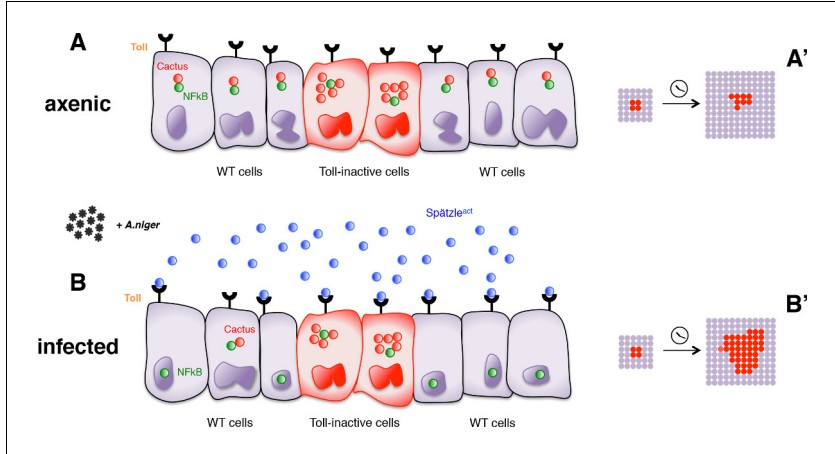

**Figure 4.** Infection-dependent role of the Toll pathway in cell growth (pathway inhibition scenario). The cartoon shows wild-type cells, colored in violet, that surround red-Cactus overexpressing cells. Toll receptors are present on cell membranes. Spätzle cytokines are drawn as blue circles, Cactus as red circles and NFkB as green circles. Infection with *A. niger* is depicted with black symbols. Nuclear shapes indicate either a dividing or a static cell (**A** and **B**). Under axenic condition, no pathogens are present in the extracellular environment and, independently of the amount of IkB inhibitor in the cytoplasm, the NFkB transcription factors are unable to translocate to the nucleus. The outcome is a leveled and uniform growth between wild-type and Cactus OE cells (**A**). When cells are naturally or artificially exposed to infectious agents, Toll pathway activation in wild-type cells causes NFkB nuclear translocation. On the contrary, Cactus OE cells remain unresponsive, with NFkB confined to the cytoplasm. This generates a strong growth advantage over the surrounding wild-type cells (**B**). After developmental time, indicated in the cartoon with a clock, clones of red Cactus OE cells are growing over a larger percentage of the tissue upon infection (**A'** and **B'**). The source of active Spätzle may be both systemic, as suggested by our experiments, and local (*Alpar et al., 2018*).

DOI: https://doi.org/10.7554/eLife.39939.014

The following figure supplement is available for figure 4:

**Figure supplement 1.** Infection-dependent role of the Toll pathway in cell growth (pathway activation scenario).
DOI: https://doi.org/10.7554/eLife.39939.015

## Materials and methods

### Drosophila stocks and care

Flies were raised at 25°C on a cornmeal food. The following lines were used in this study: *hsp-Flp, Act > CD2 > Gal4 UAS-GFP* flies were crossed with several *UAS-ORF* lines (*Bischof et al., 2013*) (*UAS-ORF* constructs are inserted in position 86Fb): *UAS-LacZ, UAS-Cactus, UAS-Toll-7, UAS-Pelle, UAS-Dorsal RNAi (VDRC), UAS-Dorsal, UAS-Toll-10b, UAS-Tollo, UAS-Toll-2* to induce clones in a wild-type background. *hsp-Flp UAS-GFP;; Rpl14⁻ SalE > Rpl14 > Gal4* and *hsp-Flp;; tub >dmyc > Gal4 UAS-GFP* flies were used to generate loser clones in combination with *UAS-LacZ, UAS-Cactus* and *UAS-RpL14*. To measure the dry weight of flies, *Act5c-Gal4/TM6b* flies were crossed with *UAS-LacZ, UAS-Cactus, UAS-Toll-10b-FLAG*. They were further crossed with *UAS-Cg1315 RNAi* (used as a control for RNAi constructs) (gift of H. Stocker), and with the VDRC line *UAS-Toll-3 RNAi*. *y w* flies were compared with *TollO⁻/⁻* (Bloomington n.1533) and *Toll-9⁻/⁻* (Bloomington n.209) mutant lines. *nub-Gal4/CyO* flies were crossed with *UAS-LacZ* and *UAS-Cactus* flies for adult wing size measurements. For eye size measurements, *ey-Flp, Act > CD2 > Gal4* flies were crossed with *UAS-LacZ, UAS-Cactus, UAS-Toll-7* and *UAS-Pelle* flies. For viability assays *Act5c-Gal4/TM6b* flies were crossed with *UAS-LacZ, UAS-Cactus, UAS-Pelle, UAS-Toll-9, UAS-Dorsal, UAS-Dif, UAS-Toll-2, UAS-Tollo, UAS-Cg1315* and with VDRC lines *UAS-Cactus RNAi, UAS-Dorsal RNAi, UAS-Dif RNAi, UAS-Toll-3 RNAi, UAS-Toll-9 RNAi*. *y w* flies were used for survival analysis and were infected 2 days after eclosion.

## Heat shock induction of loser clones and clones in a wild-type background

In the *minute* cell competition assay, a line carried a *minute* mutation in *RpL14* together with the *SalE>RpL14>Gal4* Flp-out transgene, with *SalE* being a driver expressed in the wing pouch, and *UAS-GFP*. Induction of $RpL4^{+/-}$ clones was induced via heat shock-induced Flp activity and consequent excision of the Flp-out cassette. Heat shock was induced 48 hours After Egg Deposition (AED) for 15 min. Flies were allowed to lay eggs for 8 hours. The *hsp-Flp, Act>CD2>Gal4 UAS-GFP* line was used for the induction of clones in a wild-type background. The Flp-out cassette was excised 72 h AED for 10 min by heat shock induced recombination or, alternatively, 24-36 h AED for 15 min or 48 h AED for 8 min, or 72 h AED for 6 min, as indicated in Fig. 1.

## Imaging, image analysis and quantification

Wing discs were imaged using a Leica LSM710 (25x oil or 63x oil for close-ups) confocal microscope. $GFP^+$ and $GFP^-$ areas were measured using the automatic threshold and the polygon selection tool in FIJI. In the case of the *minute* competition assay, clones were measured only in the wing pouch, in correspondence of the *SalE* expression domain. Statistical analyses were performed in Graphpad Prism 7 or Microsoft Excel. Depending on the distribution of data, t-test or Mann-Whitney test were used, unless differently indicated.

Adult eye and wing pictures were taken using an AXIO Zoom V16 Zeiss Microscope (56x magnification). Eye and wing surfaces were measured using FIJI's polygon tool.

## Immunohistochemistry

Larvae were dissected and inverted in Ringer solution, fixed in 4% PFA for 20 min at Room Temperature (RT) with rotation, washed with PBS. To label DNA, samples were stained for 10 min in 2% HINGS with 1:100 DAPI at RT with rotation and then washed in PBT/Na-Acid and PBS. For antibody stainings, samples were blocked with 2% HINGS for 30 min at RT with rotation. To detect apoptosis, samples were stained ON at 4°C with rabbit anti-cleaved-Dcp-1 primary antibody diluted 1:100 in 2% HINGS. Phospho-Histone-3 (PH3) staining was performed to detect mitotic cells. For this purpose, samples were stained ON at 4°C with rabbit anti-PH3 diluted 1:400 in 2% HINGS. Secondary antibodies were Alexa-conjugated anti-rabbit. After final washing steps in PBT/Na-Acid and PBS, wing discs were dissected in Ringer solution and mounted on slides in a drop of Vectashield Mounting Medium.

## Axenic and infected conditions

Axenic conditions. Freshly prepared cornmeal food was autoclaved together with glass vials and stoppers. Food was cooled down to about 60°C and a mix of antibiotics and antifungal agents was added to the liquid food: Pen/Strep (1.7 ml per 100 ml of food), Ampicillin (1:500), Kanamycin (1:1000), Chloramphenicol (1:1000), Antimycotic Nipagin (0.15 g per 100 ml).

About 30-50 virgin females were crossed with 20-40 males in cages and eggs were collected on sterile agar plates. Eggs were collected with sterile water, embryos were decoryonated in 10% bleach for 5 min and washed through a sieve. Embryos were then transferred into axenic food using a sterile brush. Vials were kept in a sterile incubator at 25°C.

Infected conditions. *Aspergillus niger* fungi were grown on sterile potato agar plates. For 1 liter of agar, 39 g of potato agar powder were diluted in sterile water. To avoid bacterial growth, 17 ml of Pen/Strep, 2 ml of Ampicillin, 1 ml of Kanamycin and 1 ml of Chloramphenicol were added to the mixture. Food was covered with a layer of agar covered with *A.niger*. Crosses were performed in "infected tubes" and larvae were raised at 25°C.

## Dry weight measurements

Flies were collected 14 days AED (2–3 days old, not virgins) while asleep in Eppendorf tubes and cooked for 1 hr at 95°C in an Eppendorf thermomixer compact. Dry weight was measured using a Mettler Toledo MX5 balance.

## Adult eye and wing size estimation

For eye size measurements, flies were selected and frozen for 20 min at $-20°C$. They were subsequently laid onto an agar plate with one eye oriented upwards. For wing size measurements, wings were removed with forceps from flies 12 days and placed them onto a slide in a drop of Euparal mounting medium.

## Survival analysis

After eclosion from pupal stage, 20 to 30 flies were infected with *E. coli* or *Aspergillus niger* or with PBS, the last serving as a non-infectious control solution. Flies were anesthetized with $CO_2$ and gently picked into the sternopleuron using a sterile needle (0.5 × 1.6 mm), which has been dipped into the respective solutions. Flies were then transferred into vials containing fresh food.

*E.coli* bacteria were grown ON at 37°C by inoculating 5 ml of liquid microbial growth medium with 5 µl of bacteria. 1 ml of the suspension was centrifuged at maximum speed for 3 min. The supernatant was discarded. For the infection the needle was dipped into the pellet.

*A. niger* spores were collected 5 to 7 days after plating. Spores were collected with sterile water and separated from the hyphal bodies using cotton wool filters plugged into Pasteur pipettes. The suspension was filtered five times, collected in an Eppendorf tube and centrifuged at maximum speed for 3 min. The supernatant was discarded and the procedure was repeated. The pellet of two filtration passages was resuspended in 50 µl of sterile water.

The number of living and dead flies was recorded every day. Flies were transferred to fresh food every 3 days.

## Acknowledgments

We thank Hugo Stocker, Christian Lehner, George Hausmann and Claudio Cantu' for fruitful discussion and support. FG has been partially supported by the Candoc Forschungskredit Research Grant. EM and DH were supported by SNF and ERC grants.

## Additional information

### Funding

| Funder | Author |
| --- | --- |
| Forschungskredit Candoc University of Zurich | Federico Germani |
| Swiss National Science Foundation | Daniel Hain<br>Eduardo Moreno |
| European Research Council | Daniel Hain<br>Eduardo Moreno |

The funders had no role in study design, data collection and interpretation, or the decision to submit the work for publication.

### Author contributions

Federico Germani, Conceptualization, Resources, Data curation, Software, Formal analysis, Validation, Investigation, Visualization, Methodology, Writing—original draft, Writing—review and editing; Daniel Hain, Conceptualization, Data curation, Software, Formal analysis, Validation, Investigation, Visualization, Methodology; Denise Sternlicht, Data curation, Software, Investigation, Methodology; Eduardo Moreno, Konrad Basler, Conceptualization, Supervision, Funding acquisition, Validation

### Author ORCIDs

Federico Germani (iD) https://orcid.org/0000-0002-5604-0437
Eduardo Moreno (iD) https://orcid.org/0000-0001-5040-452X

### Decision letter and Author response

Decision letter https://doi.org/10.7554/eLife.39939.018

Author response https://doi.org/10.7554/eLife.39939.019

## Additional files

### Supplementary files
• Transparent reporting form
DOI: https://doi.org/10.7554/eLife.39939.016

### Data availability
All data generated or analysed during this study are included in the manuscript and supporting files.

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
