## [Decision Letter]

Thank you for submitting your work entitled "The Toll pathway inhibits tissue growth and regulates cell fitness in an infection-dependent manner" for consideration at *eLife*. Your article has been favorably evaluated by Marianne Bronner (Senior Editor), Utpal Banerjee (Reviewing Editor), and two reviewers.

The reviewers have discussed the reviews with one another and the Reviewing Editor has drafted this decision to help you prepare a revised submission.

*Reviewer 1:–*

The present manuscript is written in a way that it suggests that the role of the Toll pathway in cell competition relies only on systemic infection. The manuscript of Laura Johnston presents clear experimental evidence that Spätzle and the proteases mediating the cleavage of Spätzle are indeed cell autonomously induced by dMyc-driven supercompetitor cells and that Spätzle expression in the wing disc is required for dMyc-driven competition.

I am just requesting that the authors acknowledge in the Abstract, Introduction and Discussion section the dual (systemic and cell autonomous) role of the Toll pathway in mediating cell competition, and, as such, revise their final model and cartoon in which they do include the data from the Johnston lab. It is fair for the cell competition field as it integrates independent experimental data coming from three different labs into a unique model.

*Reviewer 2:–*

I agree that the authors should discuss their model of the infection-dependent Toll pathway role in cell competition in light of the recently published Johnston's paper.

I think this manuscript could be suitable for *eLife* if the authors address our specific concerns.

---

## [Author Response]

Reviewer 1:The present manuscript is written in a way that it suggests that the role of the Toll pathway in cell competition relies only on systemic infection. The manuscript of Laura Johnston presents clear experimental evidence that Spätzle and the proteases mediating the cleavage of Spätzle are indeed cell autonomously induced by dMyc-driven supercompetitor cells and that Spätzle expression in the wing disc is required for dMyc-driven competition.I am just requesting that the authors acknowledge in the Abstract, Introduction and Discussion section the dual (systemic and cell autonomous) role of the Toll pathway in mediating cell competition, and, as such, revise their final model and cartoon in which they do include the data from the Johnston lab. It is fair for the cell competition field as it integrates independent experimental data coming from three different labs into a unique model.Reviewer 2:I agree that the authors should discuss their model of the infection-dependent Toll pathway role in cell competition in light of the recently published Johnston's paper.I think this manuscript could be suitable for eLife if the authors address our specific concerns.

We have modified the text to accommodate reviewers’ comments. We agree that the local production of Spätzle and the Serine proteases responsible for its activation could be triggered both locally and systemically. We therefore referenced the recent work from Laura Johnston’s lab (Alpar et al., 2018). We acknowledge their work in the Abstract, Introduction and final model. We also discuss how and why their data are possibly in line with our findings.